# The Goto Kakizaki rat: Impact of age upon changes in cardiac and renal structure, function

Patrick Meagher[1,2‡], Robert Civitarese[1‡], Xavier Lee[1,2], Mark Gordon[1], Antoinette Bugyei-Twum[1,2], Jean-Francois Desjardins[1], Golam Kabir[1], Yanling Zhang[1], Hari Kosanam[1], Aylin Visram[1,2], Howard Leong-Poi[1,3], Andrew Advani[1,3], Kim A. Connelly[1,2,3]*

1 St. Michael's Hospital, Keenan Research Centre, Li Ka Shing Knowledge Institute, Toronto, Canada, 2 Department of Physiology, University of Toronto, Toronto, Canada, 3 Institute of Medical Science, University of Toronto, Toronto, Canada

‡ These authors are co-first authors on this work.
* ConnellyK@smh.ca

## Abstract

### Background

Patients with diabetes are at a high risk for developing cardiac dysfunction in the absence of coronary artery disease or hypertension, a condition known as diabetic cardiomyopathy. Contributing to heart failure is the presence of diabetic kidney disease. The Goto-Kakizaki (GK) rat is a non-obese, non-hypertensive model of type 2 diabetes that, like humans, shares a susceptibility locus on chromosome 10. Herein, we perform a detailed analysis of cardio-renal remodeling and response to renin angiotensin system blockade in GK rats to ascertain the validity of this model for further insights into disease pathogenesis.

### Methods

Study 1: Male GK rats along with age matched Wistar control animals underwent longitudinal assessment of cardiac and renal function for 32 weeks (total age 48 weeks). Animals underwent regular echocardiography every 4 weeks and at sacrifice, early (~24 weeks) and late (~48 weeks) timepoints, along with pressure volume loop analysis. Histological and molecular characteristics were determined using standard techniques. Study 2: the effect of renin angiotensin system (RAS) blockade upon cardiac and renal function was assessed in GK rats. Finally, proteomic studies were conducted *in vivo* and *in vitro* to identify novel pathways involved in remodeling responses.

### Results

GK rats developed hyperglycaemia by 12 weeks of age (p<0.01 c/w Wistar controls). Echocardiographic assessment of cardiac function demonstrated preserved systolic function by 48 weeks of age. Invasive studies demonstrated left ventricular hypertrophy, pulmonary congestion and impaired diastolic function. Renal function was preserved with evidence of

**Data Availability Statement:** All relevant data are within the manuscript and its Supporting information files.

**Funding:** This study was funded in part by the Heart and Stroke Foundation of Canada in the form of a grant (G-15-0009282) and by St. Michael's Hospital Foundation "SCAR WARS" program in the form of funds, both to KAC. The Department of Medicine, University of Toronto also provided support in the form of a Merit Award for KAC. The specific roles of these authors are articulated in the 'author contributions' section.

**Competing interests:** The authors have read the journal's policy and have the following competing interests: KAC has received research grants to his institution from Astra Zeneca and Boehringer Ingelheim, received support for travel to scientific meeting from Boehringer Ingelheim, and received honoraria for speaking engagements and ad hoc participation in advisory boards from Astra Zeneca, Boehringer Ingelheim, and Janssen. There are no patents, products in development or marketed products associated with this research to declare. This does not alter our adherence to PLOS ONE policies on sharing data and materials.

hyperfiltration. Cardiac histological analysis demonstrated myocyte hypertrophy ($p<0.05$) with evidence of significant interstitial fibrosis ($p<0.05$). RT qPCR demonstrated activation of the fetal gene program, consistent with cellular hypertrophy. RAS blockade resulted in a reduction blood pressure($P<0.05$) cardiac interstitial fibrosis ($p<0.05$) and activation of fetal gene program. No significant change on either systolic or diastolic function was observed, along with minimal impact upon renal structure or function. Proteomic studies demonstrated significant changes in proteins involved in oxidative phosp4horylation, mitochondrial dysfunction, beta-oxidation, and PI3K/Akt signalling (all $p<0.05$). Further, similar changes were observed in both LV samples from GK rats and H9C2 cells incubated in high glucose media.

## Conclusion

By 48 weeks of age, the diabetic GK rat demonstrates evidence of preserved systolic function and impaired relaxation, along with cardiac hypertrophy, in the presence of hyperfiltration and elevated protein excretion. These findings suggest the GK rat demonstrates some, but not all features of diabetes induced "cardiorenal" syndrome. This has implications for the use of this model to assess preclinical strategies to treat cardiorenal disease.

## Introduction

Diabetes mellitus (DM) is a chronic heterogeneous metabolic disorder, characterised by hyperglycemia, which is estimated to effect 439 million adults worldwide by the year 2030 [1]. Type 2 DM (T2DM) is now disproportionately outweighing the number of type I diabetic patients (accounting for >90% of all cases), with unambiguous evidence from epidemiological and clinical data that, despite excellent glycemic control, T2DM patients have higher morbidity and mortality when compared to their non-diabetic counterparts in a range of complications such as heart failure, renal failure and death [2–4].

In particular, T2DM is associated with simultaneous impairment of both cardiac and renal function, referred to collectively as 'cardiorenal syndrome' [5]. T2DM induced changes such as impaired insulin signaling, hyperglycemia/glucotoxicity and lipotoxicity, are thought to contribute, along with hemodynamic changes, activation of the renin-angiotensin-aldosterone system (RAAS), endothelial dysfunction, inflammation and oxidative stress [6–9]. Indeed, approximately 20–40% of patients with T2D develop diabetic kidney disease, the most common cause of end stage renal failure [10].

Diabetic kidney disease is characterized by progressive glomerulosclerosis and interstitial fibrosis, in part due to over-activity of prosclerotic cytokines, such as the transforming growth factor β (TGF-β). In addition to effects on the kidneys, persons with T2DM are significant more likely to develop heart failure [11]. Despite being different organ systems, diabetic patients with worsening kidney function, as evident by reduced glomerular filtration rate and proteinuria, have a two-to-tenfold more rapid progression of cardiac related diseases, such as coronary artery disease and atherosclerosis, highlighting the interconnected nature of the cardiorenal system.

Unfortunately, despite the seminal discovery of insulin by Banting and Best in 1921, targeted treatment to prevent cardiorenal disease in T2DM individuals remain extremely limited. Until recently ACE inhibitor treatment has been the most effective treatment that improves morbidity and mortality but, the introduction of Sodium Glucose Co-Transporter 2 inhibitors

(SGLT-2i) has demonstrated benefit in cardiovascular outcomes and all-cause mortality in T2DM patients [12–14]. Further, the Canagliflozin and Renal Events in Diabetes with Established Nephropathy Clinical Evaluation (CREDENCE) trial and the Dapagliflozin and Prevention of Adverse Outcomes in Chronic Kidney Disease (DAPA-CKD) trial has recently demonstrated effectiveness to prevent kidney failure and mortality in DM patients with nephropathy [15, 16].

The major obstacle to treating T2DM in an attempt prevent cardio-renal morbidity and mortality is the lack of detailed understanding of the mechanism(s), which has significantly hindered drug development. SGLT2 inhibitors are an example of such deficits in the understanding of the mechanisms of cardiovascular complications in T2DM patients. While there has been such a significant amount of study into its action within the heart and kidney, there is still no unified hypothesis to account for the protective effects of SGLT2 inhibitors [14, 15, 17].

In order to gain a better understanding of the mechanisms via which diabetic drugs elicit their effects, a robust pre-clinical model that expresses similar cardiorenal complications as seen in T2DM patients is imperative. The Goto-Kakizaki (GK) rat is a non-obese, non-hypertensive rodent model of T2DM, which by 4–5 weeks develop glucose intolerance and later peripheral insulin resistance. Further, GK rats have demonstrated vascular smooth cell dysfunction a hallmark of T2DM [18]. The GK rat genetically recapitulates human diabetes, in that it is a polygenic model of disease and shares a susceptibility locus corresponding to a region on human chromosome 10 [19–21]. However, the detailed cardiorenal characterization of the GK rat has been poorly documented. Thus, in this study, we performed functional, structural and molecular characterization of the GK rat at early and late time points, in order to determine whether the GK rat is an appropriate model of cardiorenal diabetic complications. In addition, in order to evaluate the effectiveness of this model for the study of human therapeutics, we examined the therapeutic effect of perindopril, an Angiotensin Converting Enzyme inhibitor (ACEi). Finally, as multiple studies use cell culture assays to help explore and determine mechanisms, we looked to identify whether changes in the LV proteome of GK rats would be recapitulated in high glucose (HG) treated H9c2 cardiomyoblasts.

Our data demonstrates that the GK rodent model shows some similar characteristics to human T2DM, which was subsequently improved following treatment with an ACEi, albeit demonstrating a mild disease phenotype and response. However, a major concern for this model is the time it takes the GK animals to develop significant cardiac and renal dysfunction. As such, the applicability of using the GK rat as a model of T2DM remains limited and highlights the need for the development of alternate pre-clinical models.

## Methods

### Study 1: Characterization of the Goto Kakizaki rat at early and late timepoints

Age-matched, male Wistar and GK rats were studied across time and assessed at an early and late time point. Animals sacrificed at ~28 weeks of age were classified as the 'early', whereas animals sacrificed at ~48 weeks of age were classified as the 'late' time point. Animals were housed at constant room temperature ($21 \pm 1°C$) with a 12-hour light/dark cycle and were fed standard rat chow, formulated rodent diet designed to support growth, and water *ad libitum*. All animal studies were approved by the animal ethics committee at St Michael's Hospital, Toronto, Ontario Canada in accordance with the Guide for the Care and Use of Laboratory Animals (NIH Publication No. 85–23, revised 1996).

Diabetes was confirmed at 8 weeks of age in the GK rats. Each week, rats were weighed, and blood glucose was determined using Accu-check Advantage (Roche, Mississauga, ON,

Canada). At 28 weeks (Early) and 48 weeks (Late) of age, all animals also underwent metabolic caging and glycated hemoglobin (HbA1c) assessment for determination/confirmation of diabetes status. HbA1c was measured using A1cNow+ (Bayer, Sunnyvale, CA). Additionally, following metabolic caging urine protein: creatine levels were measured by the Department of Pathology, Toronto General Hospital, Toronto ON, Canada. Animals were then euthanized, and their heart, lungs and kidneys were excised, for physiological, histopathological and molecular analyses.

**Echocardiography.** Animals underwent echocardiography at 8 weeks of age, and subsequently every 4 weeks. All assessments were done using the Vevo® 2100 system and a MS-250 probe (VisualSonics, Ontario, CA). In brief, all animals were placed on a heating pad to maintain a body temperature of $37 \pm 1°C$ and secured in the supine position. Two dimensional long-axis images of the LV in parasternal long- and short-axis views at mid-papillary muscle level were acquired as previously described [22].

**Cardiac catheterization.** Cardiac catheterization was performed, as previously described [23]. In brief, animals were placed on a warming pad (42°C), intubated and ventilated using positive pressure. Rats were secured in a recumbent position and the right jugular vein was cannulated. Pressure was calibrated after warming the catheter (Model SPR-838 Millar instruments, Houston, TX, USA) in 0.9% NaCl at 37°C for 30 min. The right internal carotid was then identified and ligated cranially. A 2F miniaturized combined conductance catheter-micromanometer was inserted into the carotid artery to obtain aortic blood pressure, then advanced into the left ventricle until stable PV loops were obtained. Using the pressure conductance data, a range of functional parameters were then calculated (Millar analysis software LabChart 8.1).

**Heart histopathology.** Paraffin embedded sections of heart, each 4 μm thick, were examined. The accumulation of matrix was quantified on picrosirius red stained heart sections using computer-assisted image analysis (Halo, indica labs, Albuquerque, New Mexico, USA) in a blinded fashion, as previously reported [24]. The extent of cardiac myocyte hypertrophy was determined on haematoxylin and eosin stained sections, as adapted from the methods described by Kai and colleagues [25], and as we previously described [26].

**Analysis of cardiac gene expression.** Genes associated with cardiac dysfunction were assessed. In short, total RNA was isolated from homogenized cardiac tissue using trizol reagent. Total RNA (2 μg) was converted to cDNA (for a 20 μL reaction) using the high-Capacity cDNA Reverse Transcription Kit and stored at −20°C until further analysis. Measurement of the cardiac genes was relative to either *RPL13a* (study 1) or *RPL32* (study 2) using Micro-Amp® Optical 384-Well Reaction Plates with a ViiA™ 7 Real-Time polymerase chain reaction system, as previously reported [27]. Experiments were performed in triplicates, and data analysis was performed using Applied Biosystem's Comparative $C_T$ method.

**Kidney histopathology and function.** Paraffin embedded kidney section, each 4 μm thick were assessed to determine the extent of glomerulosclerosis from Periodic acid–Schiff (PAS) stained sections, as previously reported [28].

## Study 2: Impact of ACEi upon renal and cardiac structure and function

Age-matched, male Wistar and GK rats were studied. Animals were housed in the same conditions as per study 1.

As in Study 1, diabetes was confirmed at 8 weeks of age in the GK rats and, at 28 weeks of age once diabetes complications were established, animals were randomized to receive vehicle or perindopril erbumine (0.2mg/kg per day in drinking water) for 12 weeks. All animals also underwent metabolic caging and glycated hemoglobin (HbA1c) assessment at 28 and 48 weeks.

After 12 weeks of vehicle or perindopril erbumine (aged 48 weeks), animals underwent functional analysis, echocardiography every 4 weeks and cardiac catheterisation as outlined in study one. Following cardiac catheterization, animals were euthanized, and heart, lungs and kidneys were excised, for histopathological (PSR, H & E and PAS) analyses, and QT-PCR as described above.

### Study 3: Proteomics comparison of *in-vivo* and *in-vitro* models

Rat cardiomyoblast H9c2 cells were cultured in 5.5 mM glucose (NG) or 25 mM glucose (HG). After 48h, cells were harvested and separated into nuclear and cytoplasmic fractions. Left ventricular tissue was excised from 40 weeks old age matched male Wistar and GK rats. Nuclear and cytoplasmic fractions of tissue homogenates were obtained. Proteins from cell and tissue extracts were precipitated and subjected to reduction and alkylation, followed by trypsin digestion. Tryptic peptides were desalted and injected onto a nano-liquid chromatography system connected online to an orbitrap mass spectrometer. Peptides were resolved using gradient reverse phase liquid chromatography. To identify proteins, the mass spectrometry data was searched against rat and human international protein index databases using MASCOT and GPM algorithms. MASCOT identifications required at least X! Tandem identifications. Label-free quantitation was performed using spectral counting tools embedded in scaffold (Scaffold version 4.10.0, Proteome Software Inc., Portland, OR) software. Scaffold was used to validate MS/MS based peptide and protein identifications. Peptide identifications were accepted if they could be established at greater than 95.0% probability by the Peptide Prophet algorithm [29]. Protein identifications were accepted if they could be established at greater than 99.0% probability and contained at least 2 identified peptides. Protein probabilities were assigned by the Protein Prophet algorithm [30]. Proteins that contained similar peptides and could not be differentiated based on MS/MS analysis alone were grouped to satisfy the principles of parsimony. Proteins sharing significant peptide evidence were grouped into clusters.

The proteomic datasets for uniquely upregulated proteins in the HG H9c2 cell lysates, and GK rats were uploaded to Ingenuity Pathway Analysis (IPA) for core analysis of protein functional networks, interactions, and pathways (QIAGEN Inc., https://www.qiagenbioinformatics.com/products/ingenuity-pathway-analysis) [31].

**Statistical analysis.** Data are expressed as means ± SEM unless otherwise specified. Between group differences were analyzed by t-test or two-way ANOVA with Holm-Sidak post hoc test. All statistics were performed using GraphPad Prism 6 for Mac OS X (GraphPad Software Inc., San Diego, CA). A Fisher's exact test was conducted to determine the probability that pathways, biological functions, and diseases were over-represented in the protein dataset. A p value of <0.05 was regarded as statistically significant.

## Results

### Study 1: Characterisation of GK rats

Analysis of mean hemoglobin A1c (HbA1c, Table 1) demonstrated GK rats had a significantly higher HbA1c than Wistar rats at both time points and increased over time (p<0.0001). Body Weight (Table 1) was significantly reduced in GK rats compared to Wistar rats at both the early and late time points. There was a time dependent increase in body weight in both Wistar and GK rats (p = 0.0003). Tibial length (TL, Table 1) was significantly smaller in GK rats compared to Wistar rats at both time points. However, in keeping with growth the TL increased in both Wistar and GK rats' (p<0.0001) over time. Heart weight indexed to body weight (HW; BW, Table 1) was significantly increased in GK rats (p<0.0001) at both time points compared with Wistar rats, and there was a significant time dependent increase in the GK rats HW:BW

**Table 1. Morphometric data of the GK and Wistar rats at early and late timepoints.**

| | Early | | Late | |
|---|---|---|---|---|
| | **Wistar n = 6** | **GK n = 4** | **Wistar n = 6** | **GK n = 7** |
| HbA1C (%) | 4.77 ± 0.15 | 6.68 ± 0.88*** | 4.62 ± 0.34$^{\phi\phi}$ | 9.36± 0.73****,####,$^{\phi\phi\phi}$ |
| Body Weight (BW; g) | 681.00 ± 32.44 | 449.75± 18.64**** | 951.83± 104.68****,$^{\phi\phi\phi}$ | 392.57± 12.18****,#### |
| Tibia Length (mm) | 42.17 ± 0.75 | 38.13 ± 0.85**** | 45.67± 0.52****,$^{\phi\phi\phi}$ | 40.64± 0.63***,$^{\phi\phi\phi}$,#### |
| Heart weight (HW): BW (mg/g) | 2.20 ± 0.13 | 2.70 ± 0.15** | 1.91 ± 0.22*,$^{\phi\phi\phi}$ | 2.94 ± 0.16#,$^{\phi\phi}$ |
| Lung weight (LW): BW (mg/g) | 3.11 ± 0.33 | 3.18 ± 0.18 | 2.38 ± 0.36*,$^{\phi\phi}$ | 3.40 ± 0.21## |
| Left Kidney weight (KW): BW (mg/g) | 2.41 ± 0.23 | 3.12 ± 0.31* | 2.26 ± 0.52$^{\phi\phi}$ | 5.10± 0.35****,$^{\phi\phi\phi}$,### |

HbA1C (%) Hemoglobin A1C, Body Beight (BW/g), Tibal length (TL/mm), heart weight/body weight ratio (mg/g), lung weight/body weight ratio (mg/g), LKW/BW left kidney weight/body weight (mg/g) ratio. Where early = 28 weeks Late = 48 weeks GK = Goto Kakizaki rats, Wistar = Wistar rats.

\* = p<0.05, \*\* = p<0.01, \*\*\* = p<0.001 and \*\*\*\* = p<0.0001 when compared to early Wistar;

# = p<0.05, ## = p<0.01, ### = p<0.001 and #### = p<0.0001 when compared to late Wistar;

$\phi$ = p<0.05, $\phi\phi$ = p<0.01, $\phi\phi\phi$ = p<0.001 and $\phi\phi\phi\phi$ = p<0.0001 when compared to early GK;

A Two-Way ANOVA with multiple comparisons Holm-Sidak test was used to compare groups across different time points. Data is expressed as mean±SEM N = 4–8 per group.

(p = 0.0012). Lung weight indexed to body weight (LW:BW, Table 1) was significantly increased in GK rats compared to Wistar rats at the late time point (p<0.01) and significantly increased over time in GK rats (p = 0.0308). Kidney Weight indexed to body weight (KW:BW, Table 1) significantly increased in GK rats at both early (p<0.05) and late (p<0.001) time points compared to Wistar rats and there was a time dependent increase in both Wistar and GK rats (p<0.0001).

**Echocardiography.** Echocardiographic analysis of cardiac function demonstrated a small reduction in cardiac function in the GK rats at each time point relative to Wistar rats, which progressed with age (all p<0.005). Left ventricular diameter in diastole was significantly reduced in GK rats compared to Wistar rats (P = 0.0332), which did not change over time. Left ventricular systolic diameter significantly increased over time (P = 0.0004) but was not significantly different between GK and Wistar rats (Fig 1).

**Cardiac catheterization.** Cardiac catheterization with a high-fidelity LV pressure manometer demonstrated that GK rats had higher systolic blood pressure compared to Wistar rats (P = 0.0495). dP/dt max was increased in GK rats compared to Wistar rats at the early time point (P<0.0001); an effect that was not observed at the late time point. Further, dP/dt min was significantly increased with age (P = 0.0001) and this effect was seen to be most pronounced in GK rats (P = 0.0326). Diastolic function assessment demonstrated that Tau significantly increased over time (P = 0.0002), this increase was seen to be most significant in aged GK rats (P = 0.0118, Fig 1).

**Heart histopathology.** Myocyte hypertrophy was significantly increased in GK rats (p = 0.025) compared to that of Wistar rats, but no effect was seen over time. Picrosirius red staining demonstrated that collagen content as measured by the portion of red stained pixels was increased in GK rats compared to that of Wistar rats (GK rats, p = 0.0144) but there was no significant effect over time (all Fig 2).

**Cardiac gene expression.** Expression of genes known to be important in cardiac remodelling were assessed in both Wistar and GK rats. Atrial natriuretic peptide (ANP) mRNA was significantly increased in GK rats (p = 0.009) compared to Wistar rats and was significantly increased over time (p = 0.028). Beta myosin heavy chain (β-MHC) expression was significantly increased in GK rats (P = 0.0064) compared to Wistar rats but this effect was seen to be

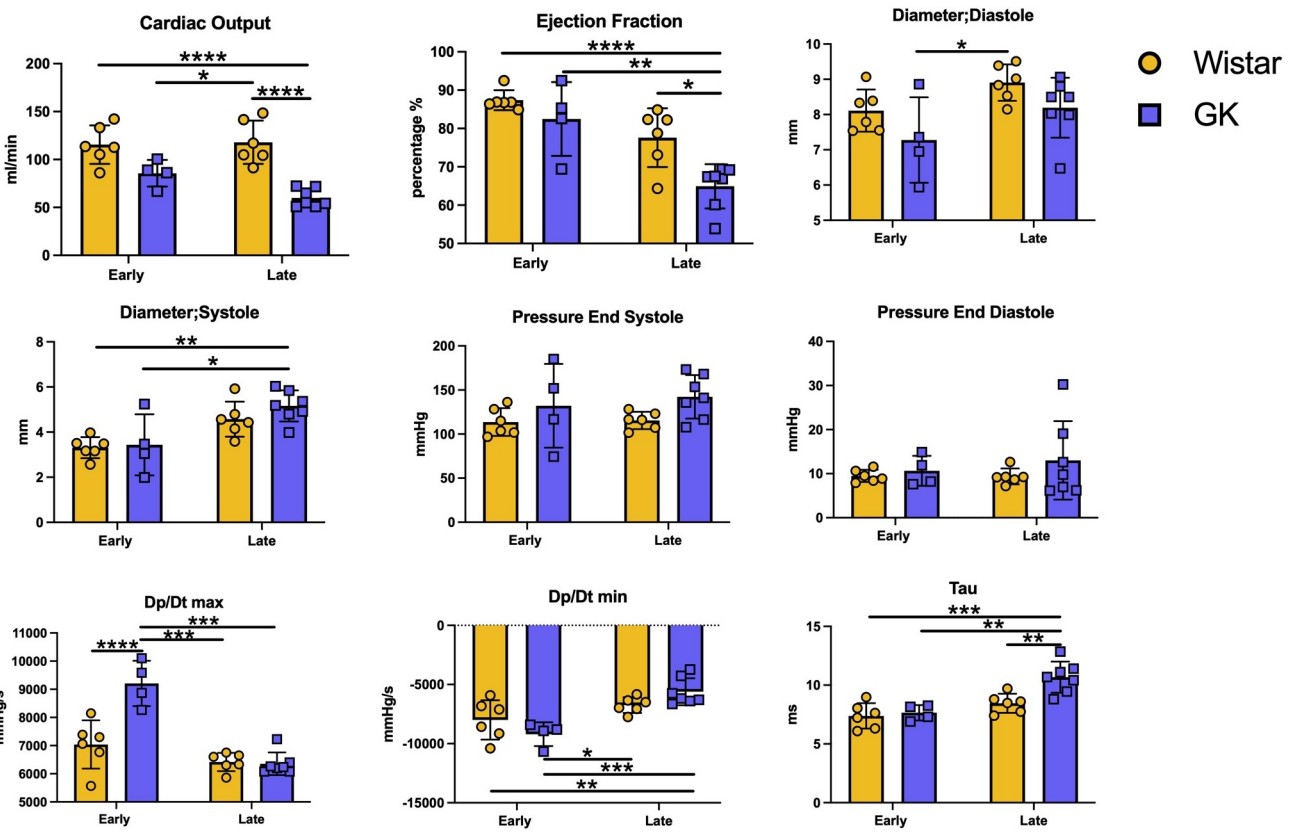

**Fig 1. Echocardiograph and cardiac catheterization parameters of Wistar and GK rats over time.** A-D represents echocardiographic analysis taken from short axis M-mode view demonstrating (A) cardiac output, (B) ejection fraction, (C) diameter in systole and (D) diameter in diastole. E-I represent analysis of cardiac catherization (E) systolic pressure, (F) end diastolic pressure, (G) Dp/dt max, (H) Dp/dt min, and (I) Tau. Where early = 28 weeks, Late = 48 weeks GK = Goto Kakizaki rats, Wistar = Wistar rats. * = p<0.05, ** = p<0.01, *** = p<0.001 and **** = p<0.0001 when comparing groups. A Two-Way ANOVA with multiple comparisons Holm-Sidak test was used to compare groups across different time points. Yellow = Wistar and Blue = GK. Data is expressed as mean±SEM N = 4–8 per group.

most prominent in the aged GK rats (p<0.0001). Alpha myosin heavy chain (α-MHC) expression was significantly increased in GK rats (p = 0.014) compared to Wistar rats and, α-MHC expression significantly increased with age (p<0.0001, Fig 3).

**Kidney histopathology and function.** The glomerulus score index (GSI) of 40 weeks old rats was significantly increased in GK rats compared to Wistar rats (1.57±0.29 vs 0.35±0.12, p<0.001, Fig 4C). Moreover, the protein:creatinine ratio was significantly increased in GK rats compared to Wistar rats (0.4±0.22 vs 0.076±0.054, p<0.05) measured at 40 weeks of age. Further, Immunohistochemical staining of Collagen IV demonstrated Collagen IV content was significantly increased in GK rats compared to Wistar rats (P<0.05) at 48 weeks (S1 Fig).

## Study 2: The effect of RAS blockade in the Goto-Kakizaki rat

**Morphology.** When GK rats were compared to aged matched GK rats treated with the ACEi perindopril, there was no significant impact upon HbA1C, body weight (BW), tibial length, heart or kidney weight (Table 2). However, the HW:BW ratio was reduced in GK rats treated with perindopril (P<0.001) compared to vehicle treated GK rats (Table 2).

**Echocardiography.** Following treatment with perindopril, echocardiography demonstrated a significant increase in cardiac output in GK rats treated with perindopril c/w vehicle

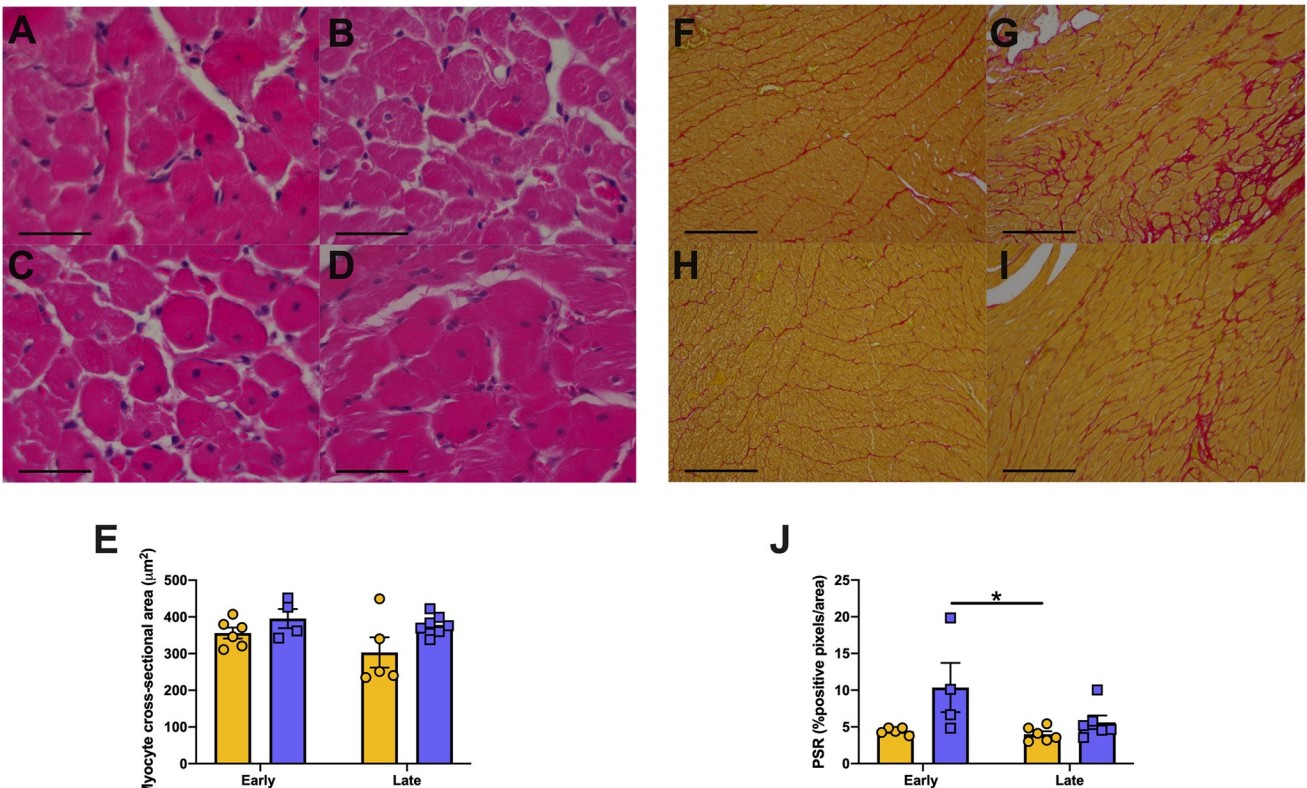

**Fig 2. Cardiac Structure in Wistar and GK rats over time.** Representative hematoxylin and eosin-stained sections and PicroSirius red stained sections. Where hearts of Wistar (A,C,F,H) rats and Goto Kakizaki (B,D,G,I) rats at early(28 weeks; A,B,F,G) and late (48 weeks; C,D,H,I) time points had myocyte hypertrophy (E) and fibrosis (J, positive PSR stained pixels) measured. * = p<0.05 when comparing groups; A Two-Way ANOVA with multiple comparisons Holm-Sidak test used to compare groups. Yellow = Wistar and Blue = GK Data is expressed as mean±SEM N = 4–8 per group. Scale bars (A-D) 50μm and (F-I) 200 μm.

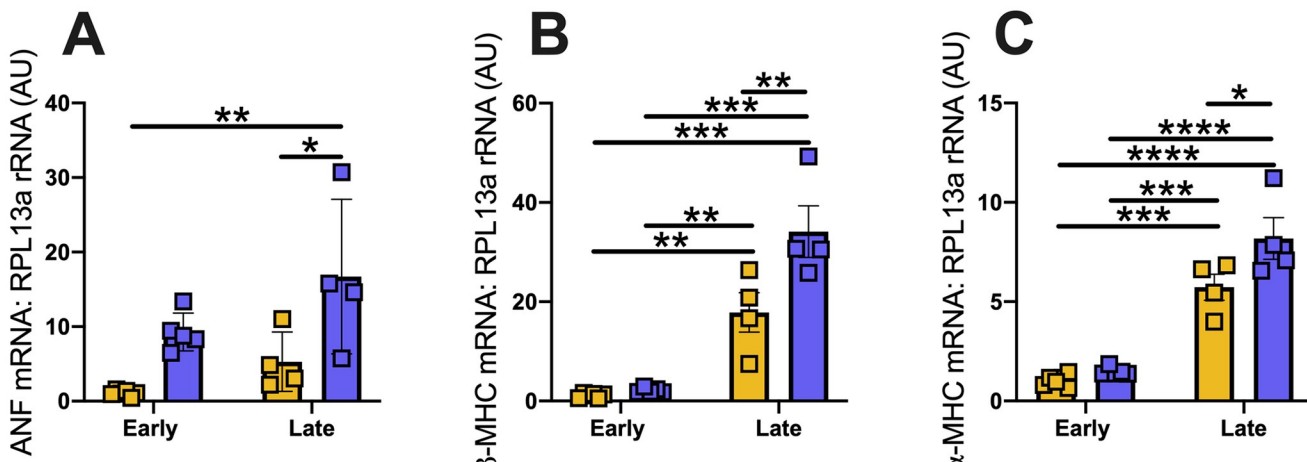

**Fig 3. RT-qPCR analysis of cardiac genes in heart tissue from GK and Wistar rats over time.** (A) Atrial natriuretic Factor, (B) beta-Myosin Heavy Chain (βMHC), (C) alpha-myosin heavy chain (αMHC). Where Early = 28 weeks and Late = 48 weeks. * = p<0.05, ** = p<0.01, *** = p<0.001 and **** = p<0.0001. A Two-Way ANOVA with multiple comparisons Holm-Sidak test was used to compare groups across different time points. Yellow = Wistar and Blue = GK. Data is expressed as mean±SEM N = 4–8 per group.

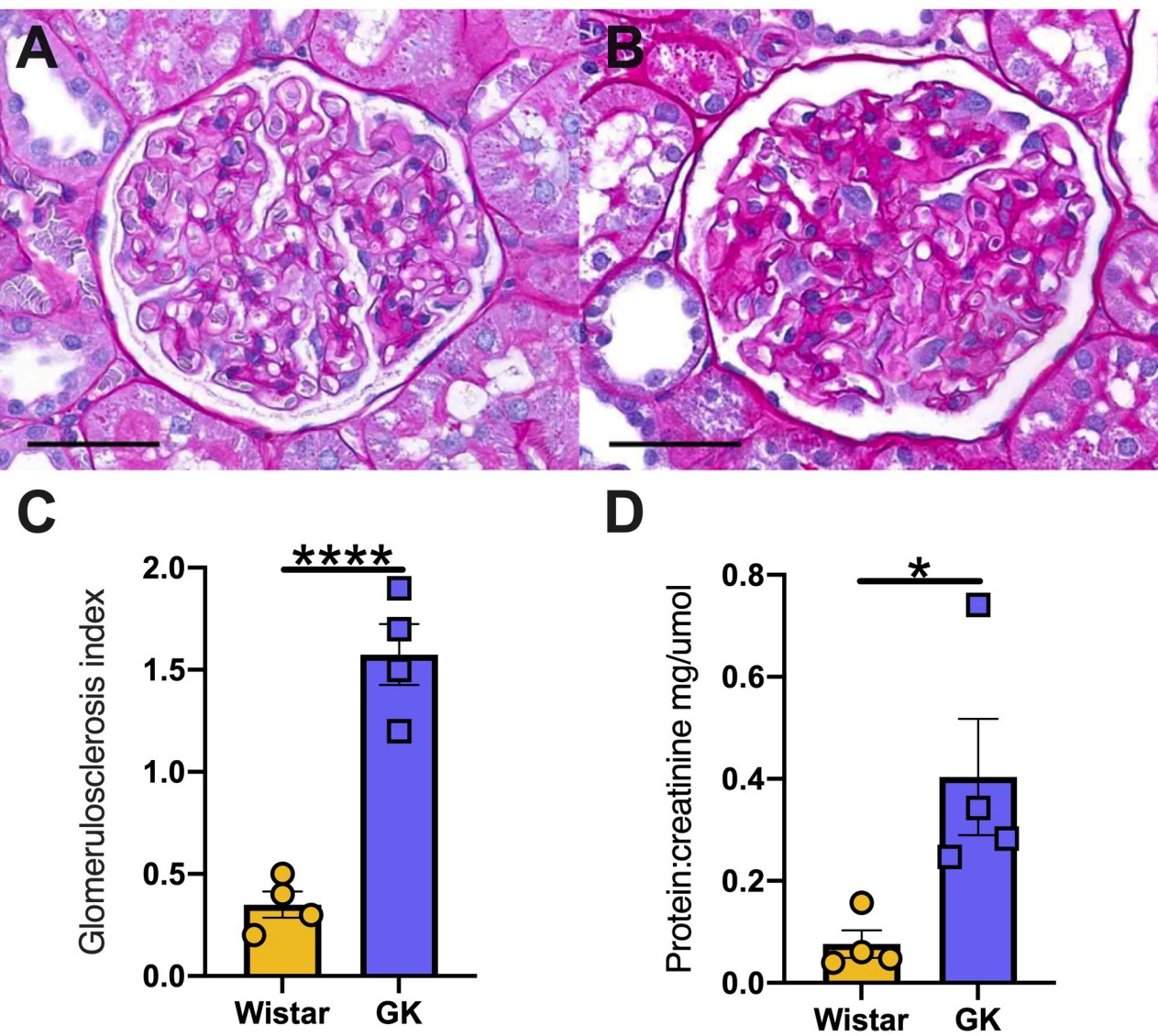

**Fig 4. Kidney structure and function in Wistar and GK at the late time point.** Representative PAS images of Wistar (A) and GK (B) kidneys at 48 weeks, respectively. Glomerulus score index (C) and analysis of protein: creatine (D) at 48 weeks of ages. * = p<0.05, **** = p<0.0001 when compared to Wistar rats; (students t-test was used to assess statistical significance). Yellow = Wistar and Blue = GK. Data is expressed as mean±SEM N = 4–8 per group. Scale bars 50 μm.

treated animals (p<0.01). Further, ejection fraction was significantly increased by perindopril treatment (p<0.05). Chamber size as assessed by internal diameter in systole or diastole was not altered by perindopril treatment (Fig 5).

**Cardiac catheterization.** Cardiac catheterization demonstrated that end systolic pressure was significantly decreased by perindopril (p<0.0001) treatment versus control GK rats. Finally, end diastolic pressure (EDP), dP/dt max, dP/dt min and Tau did not significantly change following perindopril treatment (p = N.S, Fig 5).

**Heart histopathology.** Perindopril treatment did not result in any differences in myocyte hypertrophy. Fibrosis as measured from positive PSR staining was significantly decreased following treatment with perindopril when compared to control GK rats (p<0.05, Fig 6).

**Table 2. Morphometric data of GK and GK rats following treatment with Angiotensin Converting enzyme inhibitor perindopril.**

|  | GK control (n = 4) | GK + Perindopril (n = 5) |
|---|---|---|
| HbA1C (%) | 6.03 ± 1.51 | 6.72 ± 2.43 |
| Body Weight (BW; g) | 490.25 ± 7.27 | 472.40 ± 22.40 |
| Tibia Length (mm) | 41.00 ± 0.00 | 41.00 ± 0.00 |
| Heart weight (HW): BW (mg/g) | 2.79 ± 0.10 | 2.50 ± 0.03*** |
| Lung weight (LW): BW (mg/g) | 3.14 ± 0.13 | 2.99 ± 0.25 |
| Left Kidney weight (KW): BW (mg/g) | 2.90 ± 0.23 | 2.98 ± 0.26 |

HbA1C (%); Hemoglobin A1C, Body weight;(BW;g), Tibal length;(TL/mm), HW/BW; heart weight/body weight ratio, LW/BW; lung weight/body weight ratio, LKW/BW; left kidney weight/body weight ratio (mg/g) Where *** = <0.001 and when compared to GK rats (student's t-test was used to compare groups). Data is expressed as mean ± SEM N = 4–5 per group

**Cardiac gene expression.** Following treatment with perindopril mRNA expression of ANF (p<0.05), β-MHC (p<0.001), α-MHC(p<0.001) was significantly reduced compared to control GK rats (p<0.05, Fig 7).

**Kidney histopathology and function.** 12 weeks of perindopril treatment had no impact upon GSI and protein; creatinine when compared to control GK rats (all p = N.S, Fig 8).

## Study 3: Proteomics comparison of *in-vivo* and *in-vitro* models

A total of 1191 common proteins were identified in both H9C2 cell lysates incubated in normal and high glucose media. Of these proteins, 896 were common to both HG and NG H9C2 cells. However, 128 proteins where uniquely overexpressed in HG cultures compared to that of the NG Cultures (p<0.05, Fig 9A).

Analysis of proteins from Wistar and GK rats at both the early and late time points identified 1071 common proteins. However, 253 of these identified proteins where significantly overexpressed in the GK rats compared to Wistar rats (p<0.05) (Fig 9B).

To further investigate the molecular interactions of the identified proteins, we submitted our proteomic datasets for the unique H9c2-HG and GK rat proteins for an IPA-based protein network analysis. The top-enriched networks based on a high percentage of focus proteins in our datasets included pathways linked to cardiovascular disease. cardiac hypertrophy, cancer, cell death and survival with upregulation of proteins such as MYH7, MYL2, APOA1 and phospholamban (Fig 9, Table of Top enriched networks from core analysis of both H9c2 and GK overexpressed proteins S1 Table). Commonly upregulated proteins in both H9c2 cell lysates and GK hearts were involved pathways that regulated oxidative phosphorylation, Kreb's cycle, mitochondrial dysfunction, beta-oxidation, and PI3K/Akt signalling (all p<0.05).

## Discussion

Given that cardiac and renal dysfunction remain the most common presentation of diabetes induced complications [19, 32], the structural, functional and molecular underpinnings of such derangements require elucidation. Herein we describe the time course of development of **cardiovascular** complications in a rodent model of spontaneous, non-obese type 2 DM, the Goto Kakizaki (GK) rat. We show that GK rats develop hyperglycaemia by 12 weeks of age, then progressive mild hypertension leading to left ventricular hypertrophy over 40 weeks, with evidence of impaired diastolic function, interstitial fibrosis and gene derangements over a

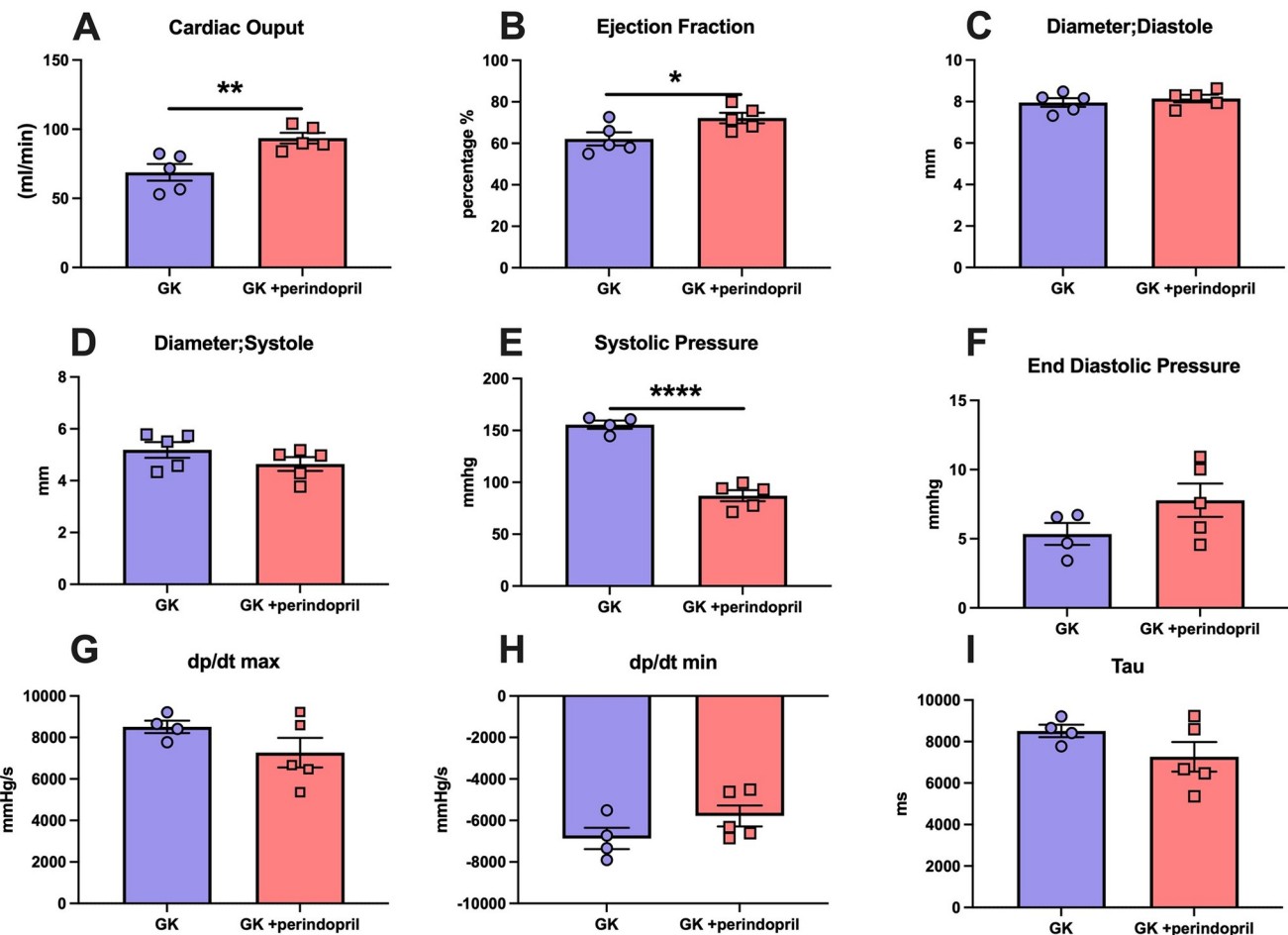

**Fig 5. Echocardiographic and cardiac catheterization parameters of GK rats and GK rat treated with Angiotensin Converting enzyme inhibitor perindopril.** A-D represents echocardiographic analysis taken from short axis M-mode view demonstrating (A) cardiac output, (B) ejection fraction, (C) diameter in systole and (D) diameter in diastole. E-I represent analysis of cardiac catherization (E) systolic pressure, (F) end diastolic pressure (G) Dp/dt max, (H) Dp/dt min and (I) Tau. All parameters were measured in Goto Kakizaki rats either treated with saline (GK) or ACE inhibitor perindopril (GK + perindopril). Where $* = p<0.05$, $** = p<0.01$, $*** = p<0.001$ and $**** = p<0.0001$ when compared to GK rats; (student's t-test was used to compare groups). Data is expressed as mean±SEM N = 4–5 per group.

28-week time period, along with the development of proteinuria. Furthermore, the GK rats demonstrate progressive and significant changes in cardiac protein expression, when compared to Wistar control rats, broadly involving alterations in contractile proteins, metabolism, hypertrophy and fibrosis.

The GK strain was developed by the successive inbreeding of Wistar rats [19, 32]. Importantly, this is a model of spontaneous, non-obese diabetes, therefore removing the confounding effects of excess adiposity upon the development of diabetes complications. Furthermore, this model shares a susceptibility locus on chromosome 10 [19] with T2 DM in humans, thus providing a genetic basis for the development of diabetes. Of note, the development of relatively mild complications is analogous to the development of complications in humans with T2DM, as this model takes 40 weeks before the development of diastolic dysfunction, and elevated urinary protein [17]. Although, it should be noted that this model is not age representative of the development of cardiorenal syndrome. In other words, it cannot be directly age matched.

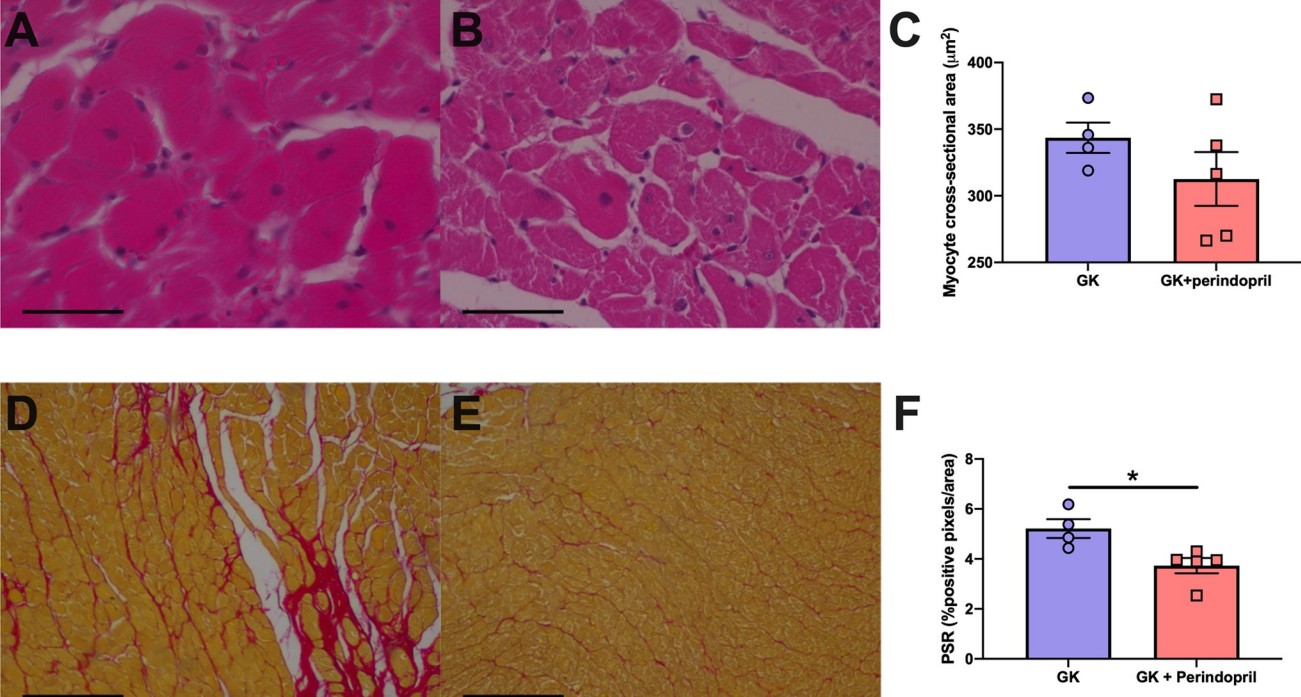

**Fig 6. Cardiac structure in control GK rats and GK rats treated perindopril.** Representative hematoxylin and eosin-stained sections and PicroSirius red stained sections. Hearts of Goto Kakizaki (A,D) rats and Goto Kakizaki rats treated with Angiotensin Converting enzyme inhibitor perindopril (B,E) had myocyte hypertrophy (C) and fibrosis (F, positive PSR stained pixels) measured. Parameters were measured in Goto Kakizaki rats either treated with saline (GK) or ACE inhibitor perindopril (GK + perindopril). * = p<0.05, when compared to GK rats (student's t-test was used to compare groups). Data is expressed as mean±SEM N = 4–5 per group. Scale bars (A-B) 50μm and (C-D) 300 μm.

## Cardiac dysfunction

Despite evidence for early left ventricular hypertrophy, systolic function was preserved, albeit at a lower number, along with evidence for abnormalities of early, active energy dependent relaxation. In order to assess systolic and diastolic function, we utilized echocardiographic parameters along with the assessment of invasive measurements of pressure. Between the early and late time points, there was a small, but significant reduction in systolic function,

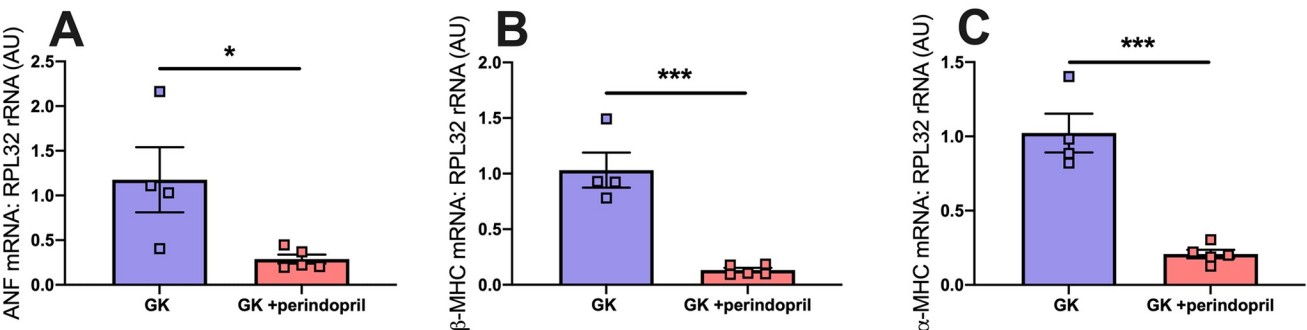

**Fig 7. RT-qPCR analysis of cardiac genes in control GK and GK rats treated with perindopril.** (A) Atrial natriuretic Factor, (B) beta-Myosin Heavy Chain (β-MHC), (C) alpha-myosin heavy chain (α-MHC). Where hearts tissue of Goto Kakizaki rats either treated with saline (GK) or ACE inhibitor perindopril (GK + perindopril) were analysed. Where * = p<0.05, when compared to GK (students t-test was used to assess statistical significance).

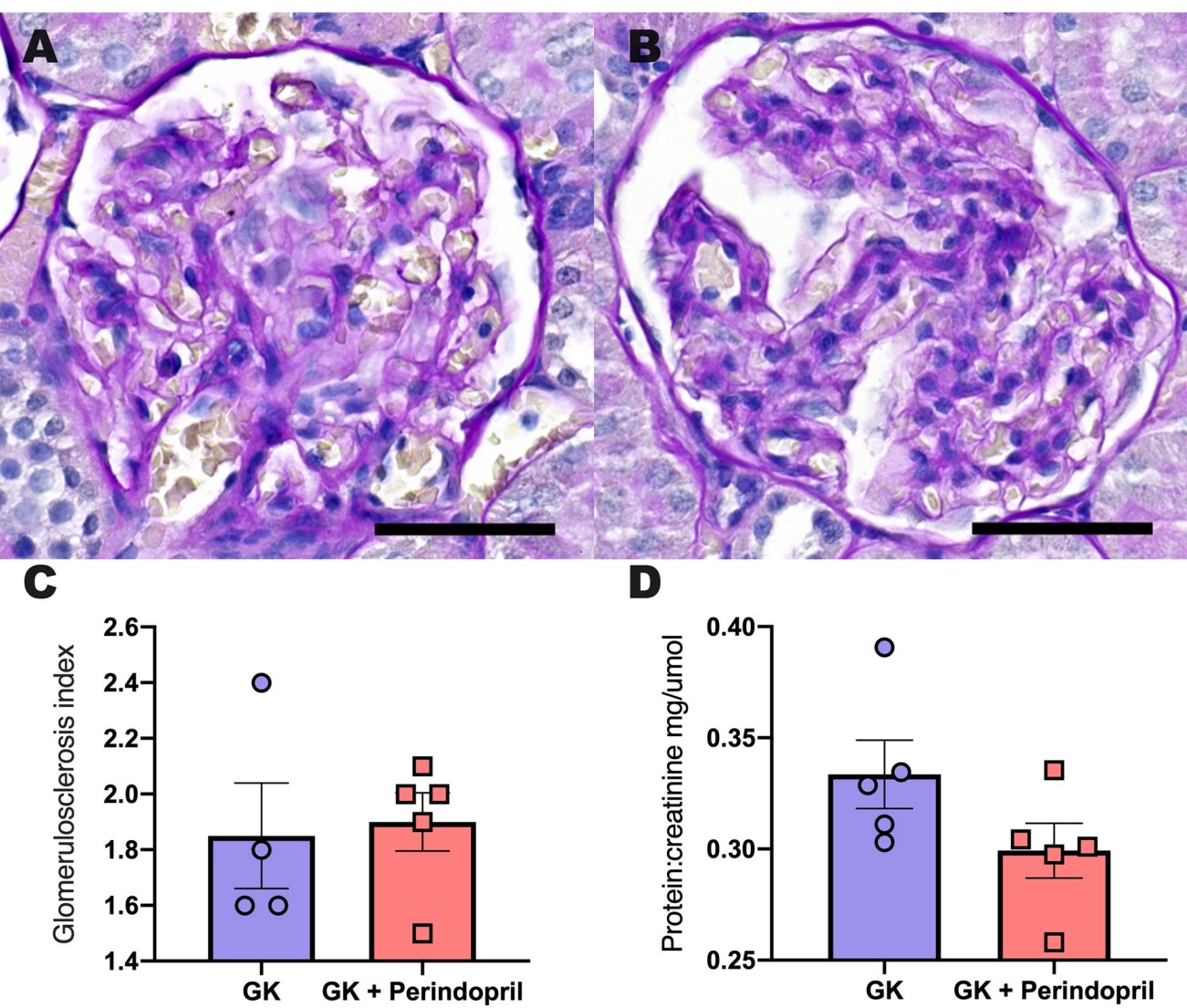

**Fig 8. Kidney Structure and function in control GK rats and GK rats treated with perindopril.** Representative PAS images of and Goto Kakizaki rats (A) and Goto Kakizaki rats treated with Angiotensin Converting enzyme inhibitor perindopril (B), respectively. Glomerulus score index (C) and analysis of protein; creatine (D) following treatment. Parameters were measured in Goto Kakizaki rats either treated with saline (GK) or ACE inhibitor perindopril (GK + perindopril). a student's t-test was used to compare groups and no significant difference was identified. Data is expressed as mean ± SEM N = 4–5 per group. Scale bars 50 μm.

ventricular volumes and LVH increased and GK rats developed an increase in lung weight. The increase in lung weight coincided with a small increase in end diastolic pressure, suggesting "pulmonary" congestion. Left ventricular hypertrophy remains a powerful predictor of adverse CV outcomes [33], with studies consistently showing regression with the use of ACEi and SGLT2i associate with improved outcomes [17, 34]. Furthermore, the findings of pulmonary congestion, subtly reduced systolic function, elevated filling pressures and LVH suggest this model, at the late stage, has features consistent with the syndrome of heart failure with preserved ejection fraction (HFpEF) [7, 35–37]. This is particularly important given the lack of evidence-based therapies to treat HFpEF.

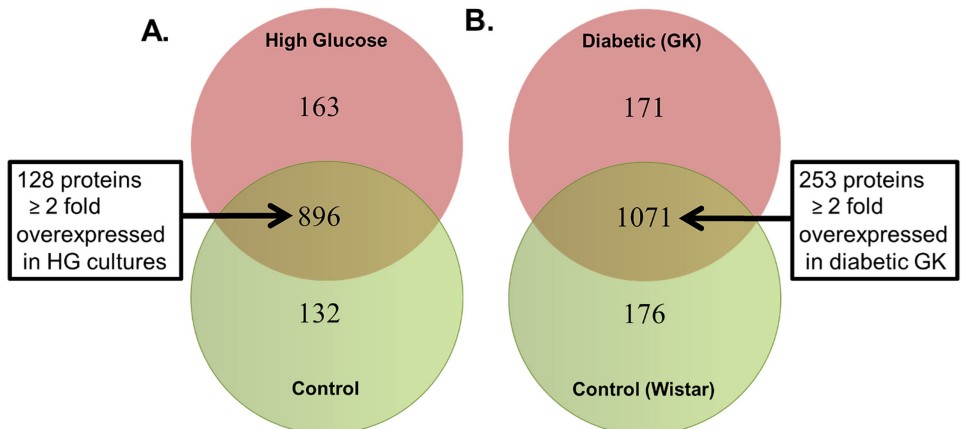

**Fig 9. Proteomic comparison of Wistar rats, GK rats and the H9C2 cell line.** (a) Commonly and exclusively quantified proteins in H9C2 cells fed either Normal Glucose or High Glucose concentration. (b) Commonly and exclusively quantified proteins in cardiac tissue from GK or Wistar rats.

Importantly, these functional findings were mirrored by the development of myocyte hypertrophy and interstitial fibrosis. Furthermore, a proteomic analysis demonstrated broad reprogramming which occurred in a time dependent fashion. Key reprogramming was seen in proteins involved in cardiac muscle contraction, glycolysis, oxidation phosphorylation, cell matrix adhesion and protein folding/sorting. This is consistent with human biopsy studies [38] whereby changes in myocyte stiffness and extracellular matrix appear responsible for some, but not all changes in cardiac function parameters. Indeed, the presence of LVH, interstitial fibrosis and altered metabolism is a prominent feature of so called diabetes induced cardiac dysfunction [39]. Of note, rodents remain resistant to the development of coronary artery disease (CAD) [34, 40] hence these findings represent the direct impact on ventricular structure and function independent on CAD.

Importantly, these findings fulfill the criteria as set out by the AMDCC [41], recapitulating many of the features of humans with T2DM, thus providing relevance. Furthermore, our findings are consistent with the reported literature. Al Kury et al provide a detailed review of cardiac function, along with identified molecular derangements in the GK rats across multiple time points [42]. The results demonstrated that at 6 months of age, GK rats had significantly lower CO and EF compared to controls. This corresponds with our evidence that GK rats had reduced cardiac function when compared to control Wistar rats. However, the GK rats in our study exhibited milder cardiac dysfunction than in the study by Al Kury et al. [42].

## Diabetic kidney disease

We demonstrated glomerulosclerosis and fibrosis accompanied by the development of mild proteinuria. Elevated urinary protein is a key diagnostic feature of the development of diabetic kidney disease, and a powerful marker for adverse CV outcome [43]. Of note, the changes were relatively mild by this late time point. The elevated urinary protein was accompanied by the typical change of diabetic nephropathy, as evidence by elevated glomerulosclerotic index. This index assesses the presence of excess extracellular matrix and mesangial expansion. Importantly, these findings occurred in the absence of significant hypertension, confirming a direct impact of hyperglycemia upon renal structure and function. Our current observation is in line with other observation within the literature which, suggest the GK rat has relatively

mild phenotype of diabetic kidney disease (DKD) [44, 45]. The development of the T2DN rat a sub-strain of the GK rat has shown much promise as it exhibits many of the hallmarks of clinical human DKD. However, the GK rat may be of use to understand the early changes which result in DKD and potentially contribute to cardio-renal syndrome.

## Impact of ACEi

Blockade of the renin angiotensin system has been a key therapeutic strategy in the treatment of human cardiac and renal disease, and is widely utilized [46, 47]. We therefore assessed the impact of a perindopril, an ACEi upon cardiac and renal outcomes. Importantly, we commenced RAS blockade at 28 weeks, once disease was progressive, and followed until the late time point. RAS blockade reduced BP and LVH as well as ANF, βMHC and αMHC expression with no significant impact upon cardiac systolic or diastolic function. These findings are in keeping with the response in humans [48]. Intriguingly, we saw little response in diabetic glomerulosclerosis at the late time point in perindopril treated GK rats, suggesting that the increase in GSI was an effect of hyperglycemia, as opposed to blood pressure related mechanisms.

## Proteomic comparison

*In-vitro* studies using H9c2 cells incubated in both normal and high concentrations is a model that is widely used to investigate molecular mechanisms associated with diabetes and cardiovascular complications [49–51]. The H9c2 cell line is an immortalized rat cardiomyoblast which, has been demonstrated to mimic the responses of primary cardiomyocytes [52]. Therefore, we employed proteomics to investigate if the H9c2 cell model would be an appropriate *in-vitro* model to investigate molecular mechanisms that are involved in cardiovascular complications in the GK rat. Our results demonstrate that pathways that modulate oxidative phosphorylation, Kreb's cycle, mitochondrial dysfunction, beta-oxidation, and PI3K/Akt signalling were similarly upregulated in cardiac tissue from GK rats and HG fed H9c2 cell. These results would suggest that the high glucose fed H9c2 cells are an appropriate *in-vitro* model to investigate molecular mechanisms eluded from the GK rat. Importantly, we provide access to the entire data set to encourage others to identify new mechanisms of diabetes-induced complications.

## Implications

Whilst the GK rat develops changes similar to patients who suffer from HFpEF, these remain relatively mild. Importantly, the model responded well to a guideline indicated therapy for cardiorenal disease, blockade of the renin-angiotensin system. These findings suggest that the GK rat is a reasonable model for testing novel therapeutic strategies in diabetes induced HFpEF. While these changes are mild, they are similar to that observed in patients with HFpEF [16, 45]. However, a number of limitations exist in this model. Firstly, the relatively long-time course and mild changes remains costly to study and do not indicate the impact of therapeutic strategies in more advanced disease. Secondly, the current study did not address hyperglycemia. The HbA1c was significantly elevated at ~9%, hence the impact of improving glucose control was not addressed in the current study. However, recent large clinical trials [53–55] demonstrate that despite aggressive therapy to improve plasma glucose, HbA1c remain poorly controlled around ~7.5–8% in the majority of patients, thus our findings remain generalizable. Thirdly, we did not study older GK rats (i.e. ~60 weeks of age), hence whether they develop significant LV dysfunction after 50 weeks was not determined. Finally, the Wistar controls developed excess weight gain (almost double GK rats). Future studies using "pair wise" feeding

may be necessary to regulate weight gain in the control animals and therefore provide a more appropriate control.

## Conclusions

By 48 weeks of age, the diabetic GK rat demonstrates evidence of preserved systolic function and impaired relaxation, along with cardiac hypertrophy, in the presence of hyperfiltration and elevated protein excretion. These findings suggest the GK rat demonstrates some, but not all features of diabetes induced "cardiorenal syndrome". This has implications for the use of this model to assess preclinical strategies to treat cardiorenal disease and may provide a model that best replicates the early complications of cardiorenal syndrome.

## Supporting information

**S1 Fig. Kidney fibrosis in Wistar and GK at the late time point.** Representative Collagen IV stained Wistar (A) and GK (B) kidneys at 48 weeks, respectively. Percent positivity staining (C) * = p<0.05 when compared to Wistar rats; (students t-test was used to assess statistical significance). Yellow = Wistar and Blue = GK. Data is expressed as mean±SEM N = 4–8 per group. Scale bars 100 μm.
(TIFF)

**S1 Table. Top five functional networks associated with overexpressed proteins identified in GK nuclear and cytosol proteomes.**
(DOCX)

**S2 Table. David pathway analysis demonstrating unique pathways significantly different in the GK rats Compare to Wistar rats.**
(DOCX)

**S1 Datasets. Proteomic datasets of GK v's Wistar rats and normal glucose and high glucose fed H9C2 cell line.**
(ZIP)

## Acknowledgments

We would like to thank Suzanne Advani for her help with the acquisition of representative images of GSI images for study 2.

## Author Contributions

**Conceptualization:** Howard Leong-Poi, Andrew Advani, Kim A. Connelly.

**Data curation:** Patrick Meagher, Robert Civitarese, Mark Gordon, Antoinette Bugyei-Twum, Aylin Visram, Kim A. Connelly.

**Formal analysis:** Patrick Meagher, Robert Civitarese, Xavier Lee, Mark Gordon, Antoinette Bugyei-Twum, Jean-Francois Desjardins, Aylin Visram, Kim A. Connelly.

**Funding acquisition:** Kim A. Connelly.

**Investigation:** Golam Kabir, Yanling Zhang, Hari Kosanam, Howard Leong-Poi, Kim A. Connelly.

**Methodology:** Jean-Francois Desjardins, Golam Kabir, Yanling Zhang, Hari Kosanam, Andrew Advani, Kim A. Connelly.

**Project administration:** Antoinette Bugyei-Twum, Kim A. Connelly.

**Software:** Jean-Francois Desjardins, Hari Kosanam.

**Supervision:** Patrick Meagher, Antoinette Bugyei-Twum, Kim A. Connelly.

**Visualization:** Patrick Meagher, Robert Civitarese.

**Writing – original draft:** Patrick Meagher, Robert Civitarese, Xavier Lee, Kim A. Connelly.

**Writing – review & editing:** Patrick Meagher, Jean-Francois Desjardins, Howard Leong-Poi, Andrew Advani, Kim A. Connelly.

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
