## [Decision Letter · Decision Letter 0]

8 Feb 2021

PONE-D-20-39982

The Goto Kakizaki rat: Impact of age upon changes in cardiac and renal structure, function

PLOS ONE

Dear Dr. Connelly,

Thank you for submitting your manuscript to PLOS ONE. After careful consideration, we feel that it has merit but does not fully meet PLOS ONE’s publication criteria as it currently stands. Therefore, we invite you to submit a revised version of the manuscript that addresses the points raised during the review process.

 All issues are required.

We look forward to receiving your revised manuscript.

Kind regards,

Vincenzo Lionetti, M.D., PhD

Academic Editor

PLOS ONE

Additional Editor Comments:

The authors should discuss their data in light of recent study demonstrating that ADAMTS 13 deficiency may underlie the onset of lethal arrhythmias in diabetes affecting lifespan (Diabetes. 2018 Oct;67(10):2069-2083.).

Journal Requirements:

2. As part of your revisions, please provide additional details pertaining to animal care and use: (1) all methods undertaken to minimize/prevent potential pain and distress, for instance: analgesics and anesthetics, supportive care, and so forth; (2) monitoring parameters - the clinical and behavioral criteria used to evaluate animal health and welfare; (3) a description of your sample size justification; (4) rate of mortality during the study (if applicable) and a brief description about your humane endpoints; (5) any unanticipated adverse events that took place during the study that affected animal health/well-being; and (6) method of euthanasia. Thank you for your attention to this matter.

"KAC has received research grants to his institution from Astra Zeneca and Boehringer Ingelheim, received support for travel to scientific meeting from Boehringer Ingelheim and honoraria for speaking engagements and ad hoc participation in advisory boards from Astra Zeneca, Boehringer Ingelheim and Janssen. All other authors declare that they have no competing interests"

6. Please amend your list of authors on the manuscript to ensure that each author is linked to an affiliation. Authors’ affiliations should reflect the institution where the work was done (if authors moved subsequently, you can also list the new affiliation stating “current affiliation:….” as necessary).

8. We noticed you have some minor occurrence of overlapping text with the following previous publication(s), which needs to be addressed:

- https://www.onlinecjc.ca/article/S0828-282X(11)01034-8/fulltext

The text that needs to be addressed involves the Abstract.

In your revision ensure you cite all your sources (including your own works), and rephrase any duplicated text outside the methods section. Further consideration is dependent on these concerns being addressed.

Reviewers' comments:

Reviewer's Responses to Questions

**Comments to the Author**

1. Is the manuscript technically sound, and do the data support the conclusions?

Reviewer #1: Partly

Reviewer #2: Yes

2. Has the statistical analysis been performed appropriately and rigorously? 

Reviewer #1: Yes

Reviewer #2: Yes

3. Have the authors made all data underlying the findings in their manuscript fully available?

Reviewer #1: Yes

Reviewer #2: Yes

4. Is the manuscript presented in an intelligible fashion and written in standard English?

Reviewer #1: Yes

Reviewer #2: Yes

5. Review Comments to the Author

Reviewer #1: In this study, Meagher and colleagues performed throughout analysis of diabetes progression and cardio-renal remodeling in GK rat, a non-obese, non-hypertensive model of type 2 diabetes. Both cardiac and renal phenotypes are assessed. Furthermore, the authors tested response to renin-angiotensin system blockade in GK rats and performed proteomic studies in rat cardiomyoblast H9c2 cells and Left ventricular tissue. While the current research does not test a novel hypothesis, it, nevertheless, makes an important report of disease progression in GK rats. The paper is well written and very readable with nice results. I have a few suggestions that I think will improve these studies.

Major:

1. One of the main conclusions of the manuscript is that the GK rat is not a good model to study cardiorenal phenotype. However, as described by the authors and others, this strain has a number of futures of both cardiac and renal disease progression. Therefore, it is suggested to focus on the discussion of the data and lessen this conclusion about the inapplicability of this model for this type of study.

2. Renal function, as described in Figure 4 and the text, was tested at 40 weeks age. However, as described for protocol 1, animals were sacrificed at ~48 weeks. Which one is correct?

3. It appears that kidney sections are available, and kidney weights to body weights were significantly different between groups. It would be nice if the authors check fibrosis.

4. Proteomic analysis: there is some inconsistency in the description of figure 9 and the numbers shown in the figure. As mentioned in the text, 163 proteins were uniquely overexpressed in HG cultures (Figure 9A), and 253 proteins are significantly overexpressed in GK rats (figure 9B), but boxes in the figure indicate 128 and 253 proteins, respectively. Furthermore, it should be important to compare the data in cell culture and in GK/Wistar rats to show overlap, if any. Please also describe key proteins/pathways identified by proteomic analysis rather than reference to supplemental table, which is also incomplete.

5. It would be great if the authors perform proteomic analysis of renal tissue as well. While it might be outside of the scope of this manuscript, this data will be important considering that the authors attempt to study cardiorenal interaction in GK rats.

6. The authors describe in the text (including abstract) that rats are age and sex-matched. While this is correct, it is slightly misleading since only male rats are studied. This is especially critical since there is a significant difference between the development of diabetes between male and female.

7. Progression of kidney disease of GK (compared to Wistar and T2DN rats, which were developed from GK rats) was recently described (PMID: 31566426). Earlier studies also described renal disease in GK and T2DN rats (PMID: 24319624). Consider comparing the findings and discuss if T2DN rats might be a better model to study cardiorenal phenotype.

8. Tables 1 and 2 – Include a number of rats for each group in the table.

9. Figure 2 legend (page 14). It appears that letters referencing to wrong images “Where hearts of Wistar (A,C,E,G) rats and Goto Kakizaki (B,D,F,H) rats at early(28 weeks; A,B,E,F) and late (48 weeks;C,D,G,H)….”. The same comment about figure 6 (figures 6F-J).

Minor:

1. Figure 2: A-D – Scales (lines/font) should be bigger. E and J: Describe a color for Wistar/GK rats.

2. Figure 2J. Is it statistical difference between Wistar and GK rats at 28 weeks?

3. Figure 4A and B – add scales.

4. Methods: Please describe what is a standard chow.

5. Please mention that 40-48 weeks old rat is not aged (40-50 years old???) when compared to human.

6. Introduction, second paragraph, reference #10. Please consider citing more updated statistics.

7. Page 11. SGLT trials (CREDENCE and DAPA-CKD) should not be bold.

Reviewer #2: In this study Meagher et al. have investigated whether the Goto-Kakizaki (GK) rat is a valid model to study the pathogenesis of diabetic cardiomyopathy. The authors find that by 48 weeks of age, the diabetic GK rat demonstrates evidence of preserved systolic function and impaired relaxation, along with cardiac hypertrophy, in the presence of hyperfiltration and elevated protein excretion. This important piece of evidence highlights how the GK rat partially demonstrates features that occur in human diabetes, including “cardiorenal syndrome”. The implications of this study are important as pre-clinical therapeutic strategies can be implemented on the GK rat to assess efficacy.

The reviewer offers the following minor critiques:

A recent study demonstrated that accelerated cerebral vascular injury in diabetes is associated with vascular smooth muscle cell dysfunction (PMID: 32166556). Is it known if these features are exhibited by the GK rat? Similarly, can the authors comment on the plasma levels of mitokines FGF21, GDF15, and Humanin in the GK rat? Evidence has suggested those mitokines are elevated in type II diabetes and Alzheimer's disease in comparison with healthy aging (PMID: 33131010)

6. PLOS authors have the option to publish the peer review history of their article (what does this mean?). If published, this will include your full peer review and any attached files.

Reviewer #1: No

Reviewer #2: No

---

## [Author Response · Author response to Decision Letter 0]

13 May 2021

Dear Dr. Lionetti, 

Thank you for giving us the opportunity to submit a revised draft of the manuscript “The Goto Kakizaki rat: Impact of age upon changes in cardiac and renal structure, function.” for publication in PLOS ONE. We greatly appreciate the time and effort that you and the reviewers committed to providing us with feedback. The critiques provided helped us immensely to improve the quality of our manuscript. We have incorporated the suggestions made by yourself and the reviewers. All changes and alterations are highlighted in the manuscript with track changes. Please see below a point-by-point response to the reviewers’ comments and recommended revisions, with the line numbers as to their location where applicable. Further we have uploaded additional supplementary material for the proteomics study.

Sincerely,

Patrick Meagher and Kim Connelly

 

Additional Editor Comments:

The authors should discuss their data in light of recent study demonstrating that ADAMTS 13 deficiency may underlie the onset of lethal arrhythmias in diabetes affecting lifespan (Diabetes. 2018 Oct;67(10):2069-2083.).

We thank the editor for this important comment. We politely ask that whilst this is important, it is outside the scope of the current paper as we did not explore the development of development of arrythmias but the onset of cardio-renal syndrome. We therefore humbly request that this not be included in the revised version of our paper.

Reviewer 1 

Major:

1. One of the main conclusions of the manuscript is that the GK rat is not a good model to study cardiorenal phenotype. However, as described by the authors and others, this strain has a number of futures of both cardiac and renal disease progression. Therefore, it is suggested to focus on the discussion of the data and lessen this conclusion about the inapplicability of this model for this type of study.

Thank-you for this important insight. This was not our intention and we have altered our main conclusion – see page 26, line 957-963.

By 48 weeks of age, the diabetic GK rat demonstrates evidence of preserved systolic function and impaired relaxation, along with cardiac hypertrophy, in the presence of hyperfiltration and elevated protein excretion. These findings suggest the GK rat demonstrates some, but not all features of diabetes induced “cardiorenal syndrome”. This has implications for the use of this model to assess preclinical strategies to treat cardiorenal disease and may provide a model that best replicates the early complications of cardiorenal syndrome.

2. Renal function, as described in Figure 4 and the text, was tested at 40 weeks age. However, as described for protocol 1, animals were sacrificed at ~48 weeks. Which one is correct?

Thank-you we have corrected this to 48 weeks . Page 10 Line 229

3. It appears that kidney sections are available, and kidney weights to body weights were significantly different between groups. It would be nice if the authors check fibrosis.

This is an excellent suggestion; we have analysed Collagen IV content in paraffin embedded kidney sections. We observed an increase in Collagen IV content in GK rats compared to Wistar rats. This has been incorporated in the text, page 16, line 85 and supplemental figure 1:

Further, immunohistochemical staining of Collagen IV demonstrated Collagen IV content was significantly increased in GK rats compared to Wistar rats (P<0.05) at 48 weeks (supplemental figure 1).

4. Proteomic analysis: there is some inconsistency in the description of figure 9 and the numbers shown in the figure. As mentioned in the text, 163 proteins were uniquely overexpressed in HG cultures (Figure 9A), and 253 proteins are significantly overexpressed in GK rats (figure 9B), but boxes in the figure indicate 128 and 253 proteins, respectively. Furthermore, it should be important to compare the data in cell culture and in GK/Wistar rats to show overlap, if any. Please also describe key proteins/pathways identified by proteomic analysis rather than reference to supplemental table, which is also incomplete.

Apologies we have rectified this in the abstract please see page 20 line 823

However, 128 proteins where uniquely overexpressed in HG cultures compared to that of the NG Cultures (p<0.05, Figure 9A).

Please also see page 20 line 831-836

The top-enriched networks based on a high percentage of focus proteins in our datasets included pathways linked to cardiovascular disease. cardiac hypertrophy, cancer, cell death and survival with upregulation of proteins such as MYH7, MYL2, APOA1 and Phospholamban (Figure 9, Table of Top enriched networks from core analysis of both H9c2 and GK overexpressed proteins Supplemental Table 1).

Finally, we did compare the HG H9C2 cells and GK rats to see if there was overlap in pathways that were commonly upregulated between them. Please the additional supplementary files which have our proteomics analysis within them. 

Please see page 20 line 836-838

Commonly upregulated proteins in both H9c2 cell lysates and GK hearts were involved pathways that regulated oxidative phosphorylation, Kreb’s cycle, mitochondrial dysfunction, beta-oxidation, and PI3K/Akt signalling (all p<0.05).

5. It would be great if the authors perform proteomic analysis of renal tissue as well. While it might be outside of the scope of this manuscript, this data will be important considering that the authors attempt to study cardiorenal interaction in GK rats.

This is a pertinent comment. The exploration of the kidney proteomics is outside the scope of the current study. Further, simple exploration of proteomics in kidney samples would likely have confounding results due to the differential protein expression between the medulla and cortex, owing to the complex nature of renal anatomy. Therefore, exploration of protein expression within these sections of the kidney would be extremely difficult to examine in whole kidney samples. We will pursue this in future studies, however. 

6. The authors describe in the text (including abstract) that rats are age and sex matched. While this is correct, it is slightly misleading since only male rats are studied. This is especially critical since there is a significant difference between the development of diabetes between male and female.

Apologies, we have amended this to make it clear we are studying male sex. 

See Abstract page 2 Line 33 and page 7 line 148

7. Progression of kidney disease of GK (compared to Wistar and T2DN rats, which were developed from GK rats) was recently described (PMID: 31566426). Earlier studies also described renal disease in GK and T2DN rats (PMID: 24319624). Consider comparing the findings and discuss if T2DN rats might be a better model to study cardiorenal phenotype.

Thank-you for the suggestion. We have made a comment on the T2DN rat, and provided appropriate reference to these two papers. see page 23 Line 591

Our current observation is in line with other observation within the literature which, suggest the GK rat has relatively mild phenotype of diabetic kidney disease (DKD) (PMID: 24319624, PMID: 31566426). The development of the T2DN rat, a sub-strain of the GK rat has shown much promise as it exhibits many of the hallmarks of clinical human DKD. However, the GK rat may be of use to understand the early changes which result in DKD and potentially contribute to cardio-renal syndrome.

8. Tables 1 and 2 – Include a number of rats for each group in the table.

Apologies. This has been corrected. See Table 1 - Page 12, Table 2 Page 16.

9. Figure 2 legend (page 14). It appears that letters referencing to wrong images “Where hearts of Wistar (A,C,E,G) rats and Goto Kakizaki (B,D,F,H) rats at early(28 weeks; A,B,E,F) and late (48 weeks;C,D,G,H)….“. The same comment about figure 6 (figures 6F-J).

Apologies. This has been corrected. Page 15 Line 348-353 & Page 18 Line 427-429

Minor:

1. Figure 2: A-D – Scales (lines/font) should be bigger. E and J: Describe a color for Wistar/GK rats.

Apologies. This has been corrected. Scale bars have been altered see new uploaded figure 2. The addition of a colour description has been added to figure legend.

2. Figure 2J. Is it statistical difference between Wistar and GK rats at 28 weeks?

Unfortunately, there was no significance between these groups.

3. Figure 4A and B – add scales.

Apologies. This has been corrected. Please see new figure 4.

4. Methods: Please describe what is a standard chow.

Apologies. This has been corrected. see methods, page 7 line 153. 

For more info on the rodent diet, the vivarium orders 18% protein diet which can be found here: https://www.envigo.com/rodent-natural-ingredient-2018-diets

5. Please mention that 40-48 weeks old rat is not aged (40-50 years old???) when compared to human.

Apologies. This has been corrected. see page 21 line 529.

Although, it should be noted that this model is not age representative of the development of cardiorenal syndrome. In other words, it cannot be directly age-matched.

6. Introduction, second paragraph, reference #10. Please consider citing more updated statistics. 

Apologies. This has been corrected. see line 81 in introduction

Reference: Roy et al. Working Group of the Endocrine Society of Bengal. Kidney Disease in Type 2 Diabetes Mellitus and Benefits of Sodium-Glucose Cotransporter 2 Inhibitors: A Consensus Statement. Diabetes Ther. 2020 Dec;11(12):2791-2827. doi: 10.1007/s13300-020-00921-y. Epub 2020 Oct 6. PMID: 33025397; PMCID: PMC7644753.

7. Page 11. SGLT trials (CREDENCE and DAPA-CKD) should not be bold. 

Apologies. This has been corrected. Page 5 line 104-105

Reviewer 2: 

A recent study demonstrated that accelerated cerebral vascular injury in diabetes is associated with vascular smooth muscle cell dysfunction (PMID: 32166556). Is it known if these features are exhibited by the GK rat?

This is outside the scope of the study. Unfortunately, in this study we did not investigate smooth muscle cell dysfunction. However, Sandu et al. (2000) demonstrated impaired vascular smooth muscle cell relaxation in GK rats (PMID: 11118023). There is no evidence that we are aware of that demonstrate accelerated cerebral vascular injury in the GK rat.

Similarly, can the authors comment on the plasma levels of mitokines FGF21, GDF15, and Humanin in the GK rat? Evidence has suggested those mitokines are elevated in type II diabetes and Alzheimer’s disease in comparison with healthy aging (PMID: 33131010)

This is outside the scope of our study. In this study we did not investigate the level of the mentioned mitokines within the plasma. Further, we did not see any significant changes in these mitokines within our proteomics analysis of the heart tissue. Finally, to our knowledge the plasma levels of the mentioned mitokines has not been investigated in the GK rat.

Grant requirements:

1) We note your current Financial Disclosure and Competing Interest read as follows:

“Studies were supported by research grants from the HSF Canada, and funds from the St Michaels Hospital Foundation “SCAR WARS” program. Dr Kim A Connelly is supported by a Merit Award from the Department of Medicine, University of Toronto.”

“KAC has received research grants to his institution from Astra Zeneca and Boehringer Ingelheim, received support for travel to scientific meeting from Boehringer Ingelheim and honoraria for speaking engagements and ad hoc participation in advisory boards from Astra Zeneca, Boehringer Ingelheim and Janssen. All other authors declare that they have no competing interests.”

We also note the following additional comments:

"Finally, Kim A. Connelly has received research grants to his institution from Astra Zeneca and

Boehringer Ingelheim, received support for travel to scientific meeting from Boehringer

Ingelheim and honoraria for speaking engagements and ad hoc participation in advisory boards

from Astra Zeneca, Boehringer Ingelheim and Janssen. However, this will have no-effect in

regard to data restriction and all pertinent data related to the study will be made available."

For the support regarding HSF Canada and St. Michael’s Hospital Foundation, please provide the following information:

a.) Any specific grant or award numbers associated with the support.

- The HSF grant has an award number which has been added. The other grants are generic and do not have a specific name or award number, hence this has been left blank.

b.) The names of the specific authors who received the support.

-the grant is to Dr Kim A Connelly. The other authors did not receive funding to support this particular project.

For the Merit Award from the Department of Medicine, University of Toronto, please provide the following information:

a.) Any specific grant or award numbers associated with the support.

-The Merit award does not have a specific name or award number, hence this has been left blank.

Please confirm whether there are any patents, products in development or marketed products associated with this research to declare.

Reply: There are no patents, products in development or marketed products associated with this research.

Please also confirm whether the support from Astra Zeneca, Boehringer Ingelheim, and Janssen, that is unrelated to the study, in any way alters your adherence to PLOS ONE policies on sharing data and materials.

Reply: There is no support from the aforementioned companies for this project. Dr Connelly has received monies for other projects. This is no way influences PLOS One policies and all data is freely available to be shared as there is no agreement with the above companies.

I have been in email contact with Anna Fodor - please see the email below in regards to disclosures.

PONE-D-20-39982R1

The Goto Kakizaki rat: Impact of age upon changes in cardiac and renal structure, function

PLOS ONE

Dear Dr. Connelly,

Your confirmation via email is acceptable, funding information should not appear in any areas of the manuscript, we will only publish funding information present in the Funding Statement section of the online submission form, which I have already updated based on your confirmation, so please just send back the manuscript, and I will process it back to the academic editor for review. 

Thanks a lot,

Anna Fodor

PLOS ONE

---

## [Decision Letter · Decision Letter 1]

21 May 2021

The Goto Kakizaki rat: Impact of age upon changes in cardiac and renal structure, function

PONE-D-20-39982R1

Dear Dr. Connelly,

We’re pleased to inform you that your manuscript has been judged scientifically suitable for publication and will be formally accepted for publication once it meets all outstanding technical requirements.

Kind regards,

Vincenzo Lionetti, M.D., PhD

Academic Editor

PLOS ONE

Additional Editor Comments (optional):

Reviewers' comments:

Reviewer's Responses to Questions

**Comments to the Author**

1. If the authors have adequately addressed your comments raised in a previous round of review and you feel that this manuscript is now acceptable for publication, you may indicate that here to bypass the “Comments to the Author” section, enter your conflict of interest statement in the “Confidential to Editor” section, and submit your "Accept" recommendation.

Reviewer #1: All comments have been addressed

2. Is the manuscript technically sound, and do the data support the conclusions?

Reviewer #1: Yes

3. Has the statistical analysis been performed appropriately and rigorously? 

Reviewer #1: Yes

4. Have the authors made all data underlying the findings in their manuscript fully available?

Reviewer #1: Yes

5. Is the manuscript presented in an intelligible fashion and written in standard English?

Reviewer #1: Yes

6. Review Comments to the Author

Reviewer #1: All concerns are addressed. I do not have any outstanding issues. Nice manuscript summarizing the use of GK rats for cardiorenal research.

7. PLOS authors have the option to publish the peer review history of their article (what does this mean?). If published, this will include your full peer review and any attached files.

Reviewer #1: No

---

## [Editor Report · Acceptance letter]

3 Jun 2021

PONE-D-20-39982R1 

The Goto Kakizaki rat: Impact of age upon changes in cardiac and renal structure, function. 

Dear Dr. Connelly:

I'm pleased to inform you that your manuscript has been deemed suitable for publication in PLOS ONE. Congratulations! Your manuscript is now with our production department. 

Kind regards, 

on behalf of

Prof. Vincenzo Lionetti 

Academic Editor

PLOS ONE